# Horizontal gene transfer is predicted to overcome the diversity limit of competing microbial species

Shiben Zhu [1], Juken Hong[1] & Teng Wang [1] ✉

Natural microbial ecosystems harbor substantial diversity of competing species. Explaining such diversity is challenging, because in classic theories it is extremely infeasible for a large community of competing species to stably coexist in homogeneous environments. One important aspect mostly overlooked in these theories, however, is that microbes commonly share genetic materials with their neighbors through horizontal gene transfer (HGT), which enables the dynamic change of species growth rates due to the fitness effects of the mobile genetic elements (MGEs). Here, we establish a framework of species competition by accounting for the dynamic gene flow among competing microbes. Combining theoretical derivation and numerical simulations, we show that in many conditions HGT can surprisingly overcome the biodiversity limit predicted by the classic model and allow the coexistence of many competitors, by enabling dynamic neutrality of competing species. In contrast with the static neutrality proposed by previous theories, the diversity maintained by HGT is highly stable against random perturbations of microbial fitness. Our work highlights the importance of considering gene flow when addressing fundamental ecological questions in the world of microbes and has broad implications for the design and engineering of complex microbial consortia.

Natural environments harbor a substantial diversity of competing microbes[1–3]. Explaining such diversity is challenging: in classic ecological models, it is highly infeasible for a large number of competing species to stably coexist[4–6]. Several mechanisms have been proposed to resolve this apparent paradox. For instance, the niche-based mechanism assumes the resource or space partitioning among different species, which, in essence, reduces the interspecies competition[7–9]. This explanation, however, is challenged by the coexistence of numerous competing species in homogeneous environments with a small number of niches[10,11]. Alternatively, neutral theory assumes that different species have similar fitness[12,13]. However, due to the short doubling time and large population size of microbes, even

small fitness differences can result in the fast domination of the fittest species[14,15].

The diversity of competing microbes is hard to explain from the static view of species fitness. However, many microbes are characterized by the fluid nature of their genomes due to the substantial interspecies or intraspecies flow of genetic materials mediated by horizontal gene transfer (HGT)[16–18]. For instance, ~43% of the genes in *Escherichia coli's* pan-genomes are acquired by HGT[19]. In human gut microbiome, over 22,000 genes were estimated to be mobilizable[20]. The flow of mobile genes that encode growth benefits or burdens enables a dynamic change of microbial fitness within ecological timescales, which might, in turn, impact the competition outcome[21].

[1]Key Laboratory of Quantitative Synthetic Biology, Shenzhen Institute of Synthetic Biology, Shenzhen Institutes of Advanced Technology, Chinese Academy of Sciences, Shenzhen 518055, China. ✉e-mail: t.wang1@siat.ac.cn

However, despite its conceptual importance, how HGT influences the diversity of competing species remains largely unknown.

Many studies have documented the rapid evolutionary change of species fitness on ecological timescales, which leads to the emergence of eco-evolutionary dynamics[22–26]. For instance, in sticklebacks[22,23], guppies[24], and cichlids[25,26], the eco-evolutionary feedbacks have been widely observed. A microbial species can also actively change its niche by modifying its nutrient uptake and metabolism[27,28]. The dynamic change of species fitness can have a significant influence on ecological outcomes, including species coexistence. HGT is a key mechanism that mediates the eco-evolutionary interplay in microbes[17]. However, a theoretical framework allowing the analysis of HGT's effects on microbial diversity remains lacking.

Understanding the interplay between HGT and microbial coexistence has implications in broad scenarios. For instance, bacterial resistance to antibiotics has become one pressing crisis facing human health. Although competing with each other in host environments, sensitive and resistant strains coexist stably, and the resistant strain often persists long even after the antibiotic selection has been removed[29]. Not knowing why sensitive and resistant strains coexist stably has become one major barrier to combat antibiotic resistance[30]. Unraveling the role of HGT in bacterial coexistence might provide insights for the design of therapeutic strategies. Indeed, antibiotic-resistance genes are often closely associated with mobile genetic elements (MGEs) like plasmids[31]. In microbiome engineering, synthetic microbial consortia have emerged as a promising tool for the production of valuable chemicals[32]. However, their applicability has been constrained by our limited ability to stably maintain the diversity of the designed communities[33]. Understanding how HGT influences microbial coexistence might provide opportunities to overcome this limitation.

Here, we established a framework of species competition by accounting for dynamic gene flow among microbes. We started by studying two-species systems and generalized our analysis to random communities of multiple members. Combining theoretical derivation and numerical simulations, we demonstrated that HGT could overcome the biodiversity limit predicted by the classic model and allow the coexistence of many competitors. In contrast with the neutral theory, the diversity maintained by HGT is stable against the fluctuations of species fitness. Our results underscore the fundamental role of gene flow in shaping the ecological dynamics and evolution of microbial communities.

## Results

To illustrate the basic concepts, we started with a community of two competing species ($s_1$ and $s_2$). The subsequent analysis is equally applicable to the competition between different strains of the same species. Without HGT, the population dynamics can be described by the generalized Lotka−Volterra (LV) model, which consists of two ordinary differential equations (ODEs) that account for species growth rates ($\mu_1$ and $\mu_2$), interspecies competition ($\gamma_1$ and $\gamma_2$) and the dilution rate ($D$) (Fig. 1a). Depending on the parameter values, the competition can result in two outcomes: one species being competed out or two species coexisting stably. The feasibility of this system to maintain diversity depends on the total volume of the parameter space that allows species coexistence. Given competition strengths, coexistence feasibility can be calculated by the fraction of growth rate combinations, out of all possibilities, that leads to stable coexistence (Fig. 1b). It can be theoretically demonstrated that the feasibility decreases with competition strength (Methods).

The classic LV model assumes $\mu_1$ and $\mu_2$ being constants, while HGT creates the dynamic change of the growth rates due to the fitness effects of the mobile genes. To describe gene flow in the model, we first dissected each growth rate into two components: the basal growth rates ($\mu_1^0$ and $\mu_2^0$) determined by the non-mobilizable genes, and the fitness effects ($\lambda_1$ and $\lambda_2$) of the mobilizable genes. Without loss of generality, we assumed that the two components combined multiplicatively: $\mu_1 = \mu_1^0(1 + \lambda_1)$, $\mu_2 = \mu_2^0(1 + \lambda_2)$. Here, positive $\lambda$ values

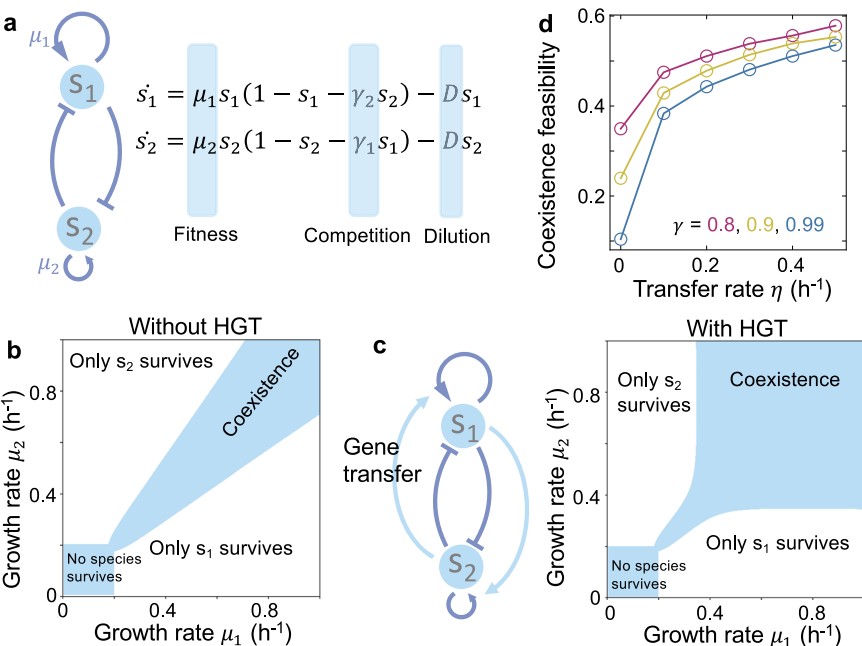

**Fig. 1 | Horizontal gene transfer (HGT) promotes the coexistence of two competing species. a** Without HGT, the competition of two species ($s_1$ and $s_2$) can be described by the LV model that consists of two ODEs. **b** The phase diagram of two competing species without HGT. The shaded areas represent the parameter regions of coexistence and no species surviving, respectively. Numerical simulations were performed with $\gamma_1 = \gamma_2 = 0.99$, $D = 0.2\,\mathrm{h}^{-1}$. **c** Gene transfer creates greater feasibility for species coexistence. Numerical simulations were performed with $\gamma_1 = \gamma_2 = 0.99$, $D = 0.2\,\mathrm{h}^{-1}$, $\eta = 0.1\,\mathrm{h}^{-1}$, $\kappa = 0.005\,\mathrm{h}^{-1}$. **d** The coexistence feasibility increases with the gene transfer rate $\eta$. For each $\eta$, we calculated the feasibility by randomizing $\mu_1$ and $\mu_2$ between 0 and 1 for 2000 times following uniform distributions. Three different values for $\gamma_1$ and $\gamma_2$ (marked with different colors) were tested.

describe fitness benefits, while negative values stand for burdens. Through gene flow, a subpopulation (denoted as $p_1$ and $p_2$) within each species acquires the mobilizable genes from its competitor. The dynamic interplay of cell growth, gene transfer rate (denoted as $\eta$), and gene loss rate (denoted as $\kappa$) determines the kinetics of $p_1$ and $p_2$, which in turn leads to the temporal change of the overall growth rates ($\mu_1^e$ and $\mu_2^e$) of each species: $\mu_1^e = \mu_1(1 + \lambda_2 \frac{p_1}{s_1})$, $\mu_2^e = \mu_2(1 + \lambda_1 \frac{p_2}{s_2})$ (see Methods for more details).

Depending on the fitness effects of the mobile genes, the dynamic change of $\mu_1^e$ and $\mu_2^e$ might alter the competition outcomes predicted by the classic model (Supplementary Fig. 1). For instance, one can imagine a scenario where a slow-growing species might evade competitive exclusion by gaining beneficial genes from its competitor (see Supplementary Fig. 1a, b for an example). To systematically quantify how gene flow impacts species coexistence, we calculated coexistence feasibility under different $\eta$, by numerical simulations with randomized parameters. Specifically, given competition strengths, we randomized $\mu_1$ and $\mu_2$ multiple times following uniform distributions while keeping $\mu_1^0$ and $\mu_2^0$ constants. Next, we simulated the population dynamics until steady states and calculated the feasibility as the fraction of growth rate combinations leading to coexistence (see Supplementary Information for more details).

Our results suggest that coexistence can be promoted by increasing $\eta$, regardless of the competition strength (Fig. 1c, d). Indeed, in the phase diagram of this community, HGT enlarges the parameter region that corresponds to stable coexistence (Fig. 1c). Here, the results are independent on the distributions of $\mu_1$ and $\mu_2$. Randomizing them following Gaussian distributions doesn't change the effects of HGT (Supplementary Fig. 2a). The assumption that $\mu_i^0$ and $\lambda_i$ combine multiplicatively is not critical, either. Calculating the growth rates by

adding $\mu_i^0$ and $\lambda_i$ ($\mu_1 = \mu_1^0 + \lambda_1, \mu_2 = \mu_2^0 + \lambda_2$) leads to the same conclusion (Supplementary Fig. 2b). The prediction is also applicable to communities with asymmetric interspecies competitions or gene transfer rates (Supplementary Fig. 2c, d). These results suggest the robustness of our analysis against a variety of confounding factors.

The model can be generalized to communities of multiple competing species. Let $m$ be the species number and $s_i$ be the abundance of the $i$-th species. $\gamma_{ij}$ represents the competition strength that the $i$-th species imposes on the $j$-th species ($i,j = 1,2,...,m$). Each species transfers its mobilizable genes (with fitness effect $\lambda_i$) to others at a rate $\eta_{ij}$. $p_{ij}$ describes the abundance of the subpopulation in the $i$-th species that acquires the mobilizable genes from the $j$-th species. The overall growth rate of the $i$-th species can be calculated based on all the mobilizable genes that it carries: $\mu_i^e = \mu_i \prod_{j \neq i}(1 + \lambda_j \frac{p_{ij}}{s_i})$. The population dynamics can then be simulated by $m + m^2$ equations that describe the temporal change of $s_i$ and $p_{ij}$, respectively (Fig. 2a, see Methods for more details).

Similarly, the coexistence feasibility can be defined as the fraction of growth rate combinations that allow the stable coexistence of all species out of all possibilities in the $m$-dimensional parameter space. We approximated the feasibility by numerical simulations with randomized $\mu_i$ values. Without gene transfer ($\eta_{ij} = 0$), feasibility decreases drastically to zero when species number increases, suggesting a maximum limit of biodiversity (Fig. 2b). Maintaining a species number over the limit is extremely unlikely in random communities. This result is consistent with the previous notion of the diversity-feasibility tradeoff in complex communities[4,6]. To understand how HGT influences diversity, we first considered gene flow in a fully connected network and assumed all species transferred the genes at the same rates. Our numerical results suggested that increasing $\eta_{ij}$ can substantially

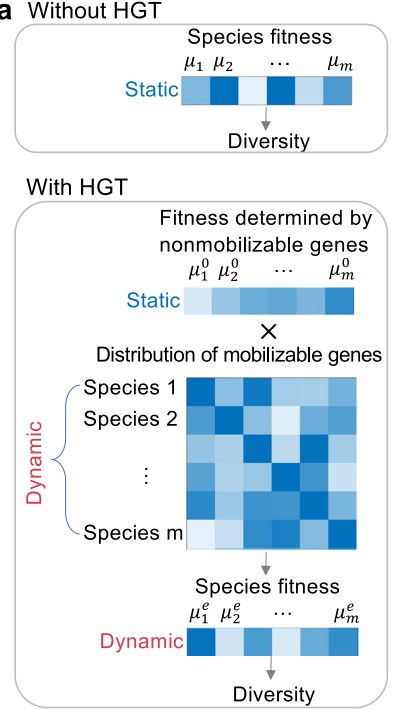

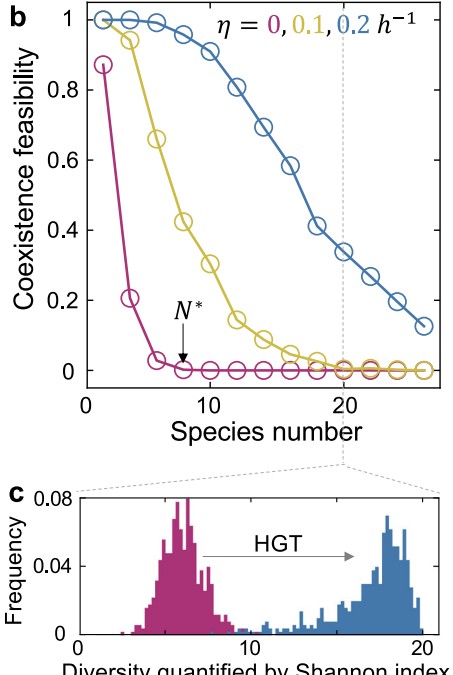

**Fig. 2 | Horizontal gene transfer (HGT) overcomes the biodiversity limit of competing species. a** A schematic of the modeling framework. Without gene flow, the diversity is determined by the static species growth rates. With gene flow, species growth rates become fluid, due to the dynamic acquisition and loss of mobile genes in each species. **b** The coexistence feasibility decreases with species number, exhibiting a biodiversity limit beyond which maintaining all the species becomes extremely infeasible. $N^*$ represents the diversity limit where the coexistence feasibility decreases to zero. Gene transfer promotes species coexistence,

making it possible for the community to overcome this limit. Coexistence feasibilities were calculated by randomizing $\mu_i$ 500 times in the range of 0.4–0.6 h⁻¹ following uniform distributions. Other parameters were $\mu_i^0 = 0.5\,h^{-1}$, $\gamma_{ij} = 0.9$, $\kappa_{ij} = 0.005\,h^{-1}$, $D = 0.2\,h^{-1}$. Three $\eta_{ij}$ values (0, 0.1 and 0.2 h⁻¹) were tested and marked with different colors. **c** Gene transfer promotes the diversity of competing species. Diversity was calculated as the Shannon index. The histogram was drawn based on the 500 communities randomly generated for each transfer rate. Two $\eta_{ij}$ values (0 and 0.2 h⁻¹) were tested and marked with different colors.

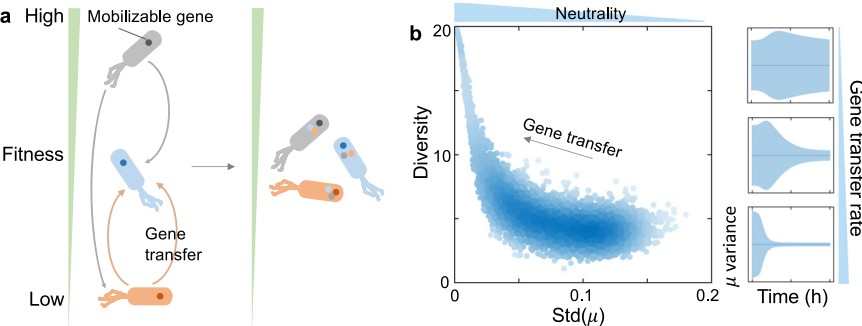

**Fig. 3 | The interpretation of HGT's effects on microbial coexistence. a** A schematic of the role of HGT. By allowing the dynamic sharing of mobilizable genes among species, HGT reduces the advantages of the strong competitors relative to the weak ones. **b** HGT promotes species coexistence and diversity by enhancing the dynamic convergence of species growth rates. In the classic LV model without HGT, population diversity decreases with the standard deviation (std) of growth rates (left). Numerical simulations were performed with the species number being 20. The species growth rates were randomized following uniform distributions around $0.5\,h^{-1}$ with random standard deviations. Other parameters are $\gamma_{ij} = 0.9$ and $D = 0.2\,h^{-1}$. HGT reduces the variations of growth rates in 20-species communities (right). Three gene transfer rates were tested ($\eta = 0.05, 0.1, 0.2\,h^{-1}$ from top to bottom). Other parameters are $\gamma_{ij} = 0.9$, $\kappa = 0.005\,h^{-1}$ and $D = 0.2\,h^{-1}$.

promote coexistence (Fig. 2b, c). In particular, for communities beyond the diversity limit predicted by the classic model, gene transfer makes the coexistence of all species possible. This prediction does not rely on the parameterization of the model and is generally applicable to partially connected networks or communities where species transfer the genes at different rates, indicating that the architecture of the transfer network is not critical for the effects of HGT (Supplementary Fig. 3).

The theoretical prediction can be intuitively understood as follows: HGT enables the coexistence of more competitors by promoting the dynamic convergence of species growth rates (Fig. 3a, b). Indeed, in communities without HGT, the steady-state diversities, in general, decrease with the standard deviation of growth rates (Fig. 3b). Species with similar growth rates are more likely to coexist against competitive exclusion. HGT allows the sharing of burdensome or beneficial genes between strong and weak competitors, which reduces the relative advantage of strong competitors and alleviates the likelihood of winner-taking-all scenarios (Fig. 3b, right panel).

In terms of promoting the fitness similarities of competing species, our interpretation of HGT's role is analogous to neutral theory, which is an extreme case where all species are assumed to have equivalent fitness. Coexistence through perfect neutrality is inherently unstable: in deterministic models, any deviations from neutrality can lead to the drastic extinction of many species[34]. In nature, however, microbial growth rates are often perturbed by the fluctuations of environmental factors such as nutrient availability[35]. To test whether gene transfer can overcome this limitation and allow robust coexistence against environmental perturbations (Fig. 4a), we carried out repeated numerical simulations by incorporating random fluctuations into species growth rates. Specifically, we dissected the population dynamics into multiple sequential intervals, each with a randomized duration. In each interval, multiple species compete with each other in a deterministic manner, and at the end of each interval, environmental fluctuations occur and cause random variations in the growth rates of all the species. We initiated each simulation with perfectly neutral populations, where all species had identical growth rates. Without gene transfer, the diversity collapses rapidly under fluctuations, confirming the instability of neutrality-mediated diversity. In contrast, HGT promotes the maintenance of diversity over a long time, suggesting that gene transfer can stabilize the competing communities in fluctuating environments (Fig. 4b–d).

## Discussion
Our work proposed an ecological mechanism of maintaining microbial diversity via gene transfer and showed the conditions where this

mechanism would potentially be effective. When mobile genes only affect species growth rates, HGT allows the stable coexistence of many competing species beyond the diversity limit predicted by the classic theory by promoting dynamic neutrality of microbial fitness. In fluctuating environments, the dynamic shuffling of mobile genes across species provides a buffering mechanism against the collapse of community diversity. Our results underscore the need to consider gene flow when studying ecological dynamics and evolution of microbial communities.

Recent studies have observed extensive diversity in microbial populations that seem to occupy similar niches. For instance, in *Vibrio* and *Synechococcus* communities, many strains within the same species can stably coexist in nearby spatial locations despite the fitness differences among these strains[10,36]. Our work provides a plausible explanation for such seemly puzzling diversity in these populations. Indeed, comparative genomics suggested that HGT within these populations is prevalent[10,36].

The empirically estimated gene transfer rates (denoted as $\eta^c$) need to be multiplied by $N_m$ before being plugged into our model (see Supplementary Information for more details). Here $N_m$ is the maximum carrying capacity of the population and has the unit of cells per mL. Therefore, the transfer rates $\eta$ in our model are several orders of magnitude higher than those measured in previous studies[37,38]. When $N_m$ is large, even slight $\eta^c$ can significantly change the coexistence feasibility. In contrast, with small $N_m$, the effects of $\eta^c$ can become negligible (Supplementary Fig. 4a). Using two empirical estimates of plasmid conjugation rates from a previous study[37], our numerical simulations suggest that the empirical HGT rates are sufficient to promote coexistence in a wide range of natural conditions[37,39] (Supplementary Fig. 4b, c). We also explored the influence of MGE fitness effects on the effective range of transfer rates. Our simulations show that when MGEs are beneficial, extremely low HGT rates can be effective in promoting diversity (Supplementary Fig. 4d; see Supplementary Information for more details). These results suggest that the role of HGT may become prominent in many environments, especially those with high cell density and beneficial MGEs. However, for burdensome MGEs in environments with very low cell densities, the contribution of HGT can be less important than other mechanisms like growth tradeoffs or cross-feeding[40,41].

Natural microbial communities are often faced with constantly varying environmental conditions that affect the ecological parameters, such as species growth rates. In order for diversity to be maintained stably, the population needs to be insensitive to the perturbations of the parameters[42–44]. Mathematically, such robustness translates into coexistence feasibility or structural stability, which

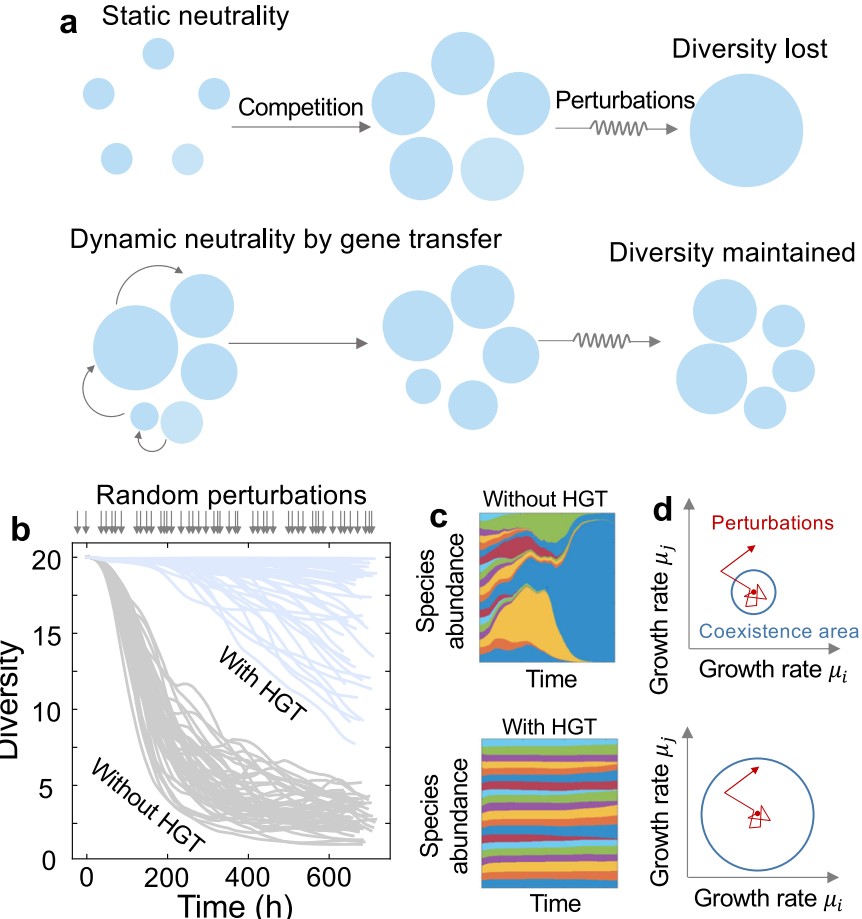

**Fig. 4 | HGT promotes the stable maintenance of species diversity in fluctuating environments. a** A schematic comparison between static neutrality in classic theory and dynamic neutrality enabled by HGT. Species coexistence by static neutrality is inherently unstable against fluctuations in species growth rates, while HGT promotes the robustness of community diversity. **b** Simulated dynamics of communities composed of 20 competing species with or without HGT. 50 repeated simulations were initiated with perfect neutrality, where all species had the same growth rates $\mu_i = 0.5\,h^{-1}$ and identical abundances. Perturbations were introduced into growth rates at random intervals. Each perturbation causes the random variations of growth rates with the magnitude of less than 5%. $\eta = 0.2\,h^{-1}$ was used as an example for dynamics with HGT. Other parameters are $D = 0.2\,h^{-1}$, $\gamma_{ij} = 0.9$, $\kappa = 0.005\,h^{-1}$. **c** Examples of population dynamics of 20 competing species without or with HGT. The widths of the colored areas represent the abundances of different species. **d** Schematic summary of HGT's role on species coexistence. By enlarging the parameter space that allows species coexistence, HGT enhances the tolerance of community diversity to growth rate fluctuations.

measures the volume of the parameter space that allows the positive abundances of all species[42–45]. We noted that structural stability, by definition, is different from dynamical stability or local asymptotic stability, which refers to the ability of a system to recover after perturbation in species relative abundances[42,43]. While the local asymptotic stability has been extensively studied in many systems[5,46,47], the determinants of structural stability have been less understood. In this work, we explicitly show that HGT might promote the structural stability of microbial communities. How HGT influences the local asymptotic stability remains an open question for future studies.

In the two-species model, we assumed that the metabolic burden or benefit of an MGE was independent of the host species or strains. However, in nature, the same MGE can have different fitness effects in different genetic backgrounds due to epistasis[48–51]. To evaluate the influence of this assumption on the conclusion, we built a model that accounted for two types of epistasis: magnitude epistasis, where the host genomic background only influences the magnitude but not the sign of the fitness effect, and sign epistasis, where the same MGE causes growth burden in one species or strain while brings fitness benefit in the other (see Supplementary Information for more details). Our numerical simulations with randomized parameters suggest that how HGT affects coexistence is dependent on the epistasis type. Magnitude epistasis does not qualitatively change the conclusion, but

sign epistasis does (Supplementary Fig. 5). When a mobile gene causes opposite fitness effects in two different genetic backgrounds, the transfer of this gene will reduce the coexistence feasibility. These results suggest that MGE epistasis might add another layer of complexities to the interplay between HGT and the coexistence of species.

Our work predicts that HGT might promote species diversity when MGEs only affect species growth rates and have no influences on inter-species competition. Certain caveats need to be considered when applying this prediction. For instance, the sharing of many mobile genes can also promote niche overlapping, leading to an increase of competition strength[52,53]. To understand how the transfer of these genes will influence species coexistence, we adapted the main model by considering the dynamic change of competition strength during gene transfer (see Supplementary Information for more details). The numerical simulations predict that when mobile genes promote inter-species competition, HGT can reduce the coexistence feasibility of competing species (Supplementary Fig. 6). These results suggest that how HGT affects species coexistence in a specific microbiota might be context-dependent. Gene transfer can promote or suppress microbial coexistence, depending on epistasis and biological traits encoded by the mobile genes.

Our results are in line with the previous studies on the relationship between HGT and microbiota stability[54,55]. For instance, one study

focused on the interaction between microbial cooperator and cheater and showed that HGT could promote the coexistence of these two genotypes[54]. Another study specifically showed that horizontal transfer of resistance genes could promote microbiome stability in response to environmental stressors[55]. While these studies only considered specific systems, our work resonates with their conclusions and generalizes the role of HGT in the broader context of microbial communities.

In our extended LV model, the dynamics of species fitness during HGT arise from the changes in population structure: in each species, HGT generates subpopulations whose growth rates differ from the others due to the metabolic burden or benefits of the mobile genes. However, dynamic fitness can also emerge from the shifts of environmental factors[56–58]. The influence of environmental changes on species coexistence has been extensively studied[59–61]. Together with these previous studies, our work highlights the importance of considering the dynamic nature of species fitness in the field of ecology.

The fitness effect of an MGE can be discrete. For instance, under strong antibiotic selections, only cells carrying antibiotic-resistant MGEs can survive. To examine whether our conclusion is still applicable in this scenario, we generalized our model by considering the transfer of an antibiotic-resistant MGE in a population of $m$ species (see Supplementary Information for more details). Our numerical result suggests that without HGT, only the donor species carrying the MGE can survive due to antibiotic selection. Increasing the HGT rate promotes species coexistence and diversity by allowing more species to be resistant to antibiotic killing (Supplementary Fig. 9). These results suggest the applicability of our conclusion to the scenario of discrete fitness effects.

In our model, we assumed the number of the MGEs equaled the species number. However, in natural systems, the MGE diversity might be higher than chromosomes due to immigration or de novo mutations[62]. To understand whether our conclusion is still applicable when the diversity of MGEs changes, we established an extended model that accounts for the flow of an arbitrary number of MGEs in a community of multiple species. By numerical simulations with randomized parameters, our results show that the coexistence feasibility increases with MGE diversity (Supplementary Fig. 10). In addition, regardless of the MGE diversity, a faster HGT rate consistently leads to a greater possibility of coexistence. These results suggest that enhancing genetic exchange among microbes, through either increasing MGE diversity or increasing HGT rate, can promote microbial diversity.

The competition between the preexisting strain and the mutants is at the core of microbial evolution[63]. The evolutionary rates are known to vary across different species, while the mechanisms shaping the evolutionary rates have been largely unknown[64]. Our work indicates that, by promoting the coexistence of different bacterial types, HGT can impede the selective sweep of the fittest strain, which might have a substantial influence on the evolutionary pace of population growth rates.

A previous study suggested that HGT prevents vertical selective sweeps when migration is present[65]. Our work shows that HGT can enable the stable coexistence of strong and weak competitors, reducing the likelihood of winner-taking-all scenarios while allowing the mobile genes to spread across species, which is in line with their results. Our results also suggest the role of HGT can be amplified by greater MGE diversity. Therefore, MGE diversity through gene flow with external populations can further promote the frequency of horizontal sweeps relative to vertical sweeps. Such MGE diversity might allow the horizontal sweeps of alleles under positive selections even in the absence of immigration.

Manipulation of microbial coexistence has important applications in different scenarios[66]. For instance, in waste treatments or fuel production, designing communities that maintain diversity is instrumental for overall efficiency and yield[67]. In the human gut, loss of microbial diversity is closely associated with many diseases[68]. In soil,

bacterial diversity is also critical to maintaining plant productivity[69]. Our work suggests that controlling gene transfer rates can potentially be an effective strategy to engineer the diversity of complex communities. For instance, introducing efficient MGEs like broad-host plasmids can promote genetic sharing among bacteria[70], while spatial partitioning or treatments of some small chemicals might remodel or block the gene transfer networks[9,71].

## Methods

### Mathematical model of two competing species

For the competition of two species without HGT, the classic Lotka–Volterra model consists of two ODEs:

$$\frac{ds_1}{dt} = \mu_1 s_1 (1 - s_1 - \gamma_2 s_2) - D s_1 \tag{1}$$

$$\frac{ds_2}{dt} = \mu_2 s_2 (1 - s_2 - \gamma_1 s_1) - D s_2 \tag{2}$$

where $\mu_1$ and $\mu_2$ are the growth rates of the two species. $\gamma_1$ and $\gamma_2$ are the competition strengths. $D$ is the dilution rate. Depending on the parameter values, the community can reach different compositions at steady states. The condition of coexistence ($s_1 > 0$ and $s_2 > 0$ at steady state) can be analytically derived as:

$$\gamma_1 < \phi_2/\phi_1 < \frac{1}{\gamma_2}$$

where $\phi_1 = \frac{\mu_1 - D}{\mu_1}$ and $\phi_2 = \frac{\mu_2 - D}{\mu_2}$. Species 2 will compete out species 1 when $\phi_2/\phi_1 > \frac{1}{\gamma_2}$, while species 1 will be the only survivor if $\gamma_1 > \phi_2/\phi_1$. Increasing competition strengths will narrow down the range of $\phi_2/\phi_1$ that allows coexistence. Thus, coexistence feasibility decreases with competition strength.

To account for HGT in the model, we dissected $\mu_1$ and $\mu_2$ into two components: the basal growth rates ($\mu_1^0$ and $\mu_2^0$), and the fitness effects ($\lambda_1$ and $\lambda_2$) of the mobilizable genes. We assumed $\mu_1 = \mu_1^0 (1 + \lambda_1)$, $\mu_2 = \mu_2^0 (1 + \lambda_2)$. HGT creates a subpopulation (denoted as $p_1$ and $p_2$) within each species that acquires the mobilizable gene from its competitor. The growth rates of $p_1$ and $p_2$ can then be obtained as $\mu_1 (1 + \lambda_2)$ and $\mu_2 (1 + \lambda_1)$, respectively. The extended LV model includes four ODEs that describe the dynamics of $s_1$, $s_2$, $p_1$, and $p_2$, respectively:

$$\frac{ds_1}{dt} = \mu_1^e s_1 (1 - s_1 - \gamma_2 s_2) - D s_1 \tag{3}$$

$$\frac{ds_2}{dt} = \mu_2^e s_2 (1 - s_2 - \gamma_1 s_1) - D s_2 \tag{4}$$

$$\frac{dp_1}{dt} = \mu_1 (1 + \lambda_2) p_1 (1 - s_1 - \gamma_2 s_2) + \eta_1 (s_2 + p_1)(s_1 - p_1) - (D + \kappa_1) p_1 \tag{5}$$

$$\frac{dp_2}{dt} = \mu_2 (1 + \lambda_1) p_2 (1 - s_2 - \gamma_1 s_1) + \eta_2 (s_1 + p_2)(s_2 - p_2) - (D + \kappa_2) p_2 \tag{6}$$

The effective growth rates $\mu_1^e$ and $\mu_2^e$ of the two species are determined by the abundances of $p_1$ and $p_2$, respectively: $\mu_1^e = \mu_1 (1 + \lambda_2 \frac{p_1}{s_1})$, $\mu_2^e = \mu_2 (1 + \lambda_1 \frac{p_2}{s_2})$. In Eqs. (5) and (6), the first terms describe the population growth of $p_1$ and $p_2$. The second terms describe the horizontal gene flow from donors ($s_2 + p_1$ and $s_1 + p_2$) to recipients ($s_1 - p_1$ and $s_2 - p_2$). $\eta_1$ and $\eta_2$ are transfer rates. $\kappa_1$ and $\kappa_2$ are the loss rates of the mobilizable genes.

Here, $\mu_1$ and $\mu_2$ are the static growth rates when interspecies gene transfer is absent. With HGT, however, the effective growth rates $\mu_1^e$ and $\mu_2^e$ become dynamic, which is dependent on the fractions of $p_1$ or $p_2$ within each species. With given values of $\mu_1$, $\mu_2$, $\gamma_1$, $\gamma_2$, $\eta_1$, $\eta_2$, $\kappa_1$, $\kappa_2$, and $D$, we can simulate the population dynamics and analyze the influence of HGT on coexistence.

## Calculating the coexistence feasibility of two competing species

To quantify how horizontal gene transfer influences the coexistence of two competing species, we calculated the coexistence feasibility under different $\eta$ values, by numerical simulations with randomized parameters. Specifically, for each given $\eta$ value, we randomized $\mu_1$ and $\mu_2$ 2000 times between 0 and $1\,h^{-1}$ following uniform distributions. The other parameters are $\mu_1^0 = \mu_2^0 = 0.5 h^{-1}$, $\kappa = 0.005 h^{-1}$, $D = 0.2 h^{-1}$. The fitness effect ($\lambda_i$) of each MGE was obtained as $\lambda_i = \frac{\mu_i}{\mu_i^0} - 1$. Each simulation was initiated with equal abundances of $s_0$ and $s_1$ ($s_0 = s_1 = 0.5$). The dynamics of each population were simulated for up to 200 h, until steady states were reached. The two species were defined as coexisting when their abundances were both larger than 0.01. The coexistence feasibilities were then calculated as the fractions of communities that ended up with coexistence out of 2000 randomized populations.

For symmetric competition, we assumed $\gamma_1 = \gamma_2$, while for asymmetric competition, we assumed $\gamma_1 = \gamma_2/2$ (Supplementary Fig. 2c). Similarly, for symmetric transfer of genes, we assumed $\eta_1 = \eta_2$ (Fig. 1d), while for asymmetric competition, we assumed $\eta_1 = \eta_2/2$ (Supplementary Fig. 2d).

## Mathematical model of multiple competing species

Our model can be readily extended to complex communities composed of multiple species. For a community of $m$ species, the model includes two groups of ODEs:

$$\frac{ds_i}{dt} = \mu_i^e s_i \left(1 - s_i - \sum_{j \neq i} \gamma_{ji} s_j\right) - D s_i \tag{7}$$

$$\frac{dp_{ij}}{dt} = \mu_i(1 + \lambda_{ij}) \left[\prod_{k \neq i,j}\left(1 + \lambda_{ik}\frac{p_{ik}}{s_i}\right)\right] p_{ij}\left(1 - s_i - \sum_{j \neq i}\gamma_{ji} s_j\right)$$
$$+ \left(s_i - p_{ij}\right)\sum_{k=1}^{m} \eta_{jki} p_{kj} - \left(D + \kappa_{ij}\right) p_{ij}.(i \neq j) \tag{8}$$

Here, $s_i$ represents the abundance of the $i$-th species, and $p_{ij}$ represents the abundance of cells in the $i$-th species that acquires $s_j$-originated mobile genes. We assumed $p_{ii} = s_i$, while for $i \neq j$, the dynamics of $p_{ij}$ is described by Eq. (8). $\mu_i^e$ is the effective growth rate of $s_i$ and can be calculated by $\mu_i^e = \mu_i \prod_{j \neq i}\left(1 + \lambda_{ij}\frac{p_{ij}}{s_i}\right)$. $\mu_i$ is defined as $\mu_i = \mu_i^0(1 + \lambda_{ij})$. $\mu_i^0$ is the basal growth rate of the $i$-th species determined by the non-mobilizable genes. $\lambda_{ij}$ is the fitness effect of the $s_j$-originated mobile genes in the $i$-th species. $\gamma_{ji}$ describes the negative interaction that $s_j$ imposes on the $i$-th species. $\eta_{jki}$ is the transfer rate of the $s_j$-originated genes from species $k$ to species $i$. $D$ and $\kappa_{ij}$ are the dilution and gene loss rate, respectively.

## Calculating the coexistence feasibility of multiple competing species

For a community of $m$ competing species, the coexistence feasibility is defined as the fraction of growth rate combinations that allow the stable coexistence of all species. For each given species number between 2 and 26, we approximated the coexistence feasibility by numerical simulations with randomized $\mu_i$ values between 0.4 and 0.6 $h^{-1}$ following uniform distributions. The fitness effect ($\lambda_{ji}$) of each MGE was calculated as $\lambda_{ji} = \frac{\mu_i}{\mu_i^0} - 1$ where $\mu_i^0 = 0.5 h^{-1}$. Here, we assumed that the fitness effect of each type of MGEs is independent of the host species. The other parameters are $\kappa_{ij} = 0.005 h^{-1}$,

$D = 0.2 h^{-1}$ and $\gamma_{ij} = 0.9 h^{-1}$. A population was defined as coexisting when the abundances of all species were larger than 0.01.

## The relationship between growth rate variability and population diversity

Numerical simulations were performed using the classis LV model without HGT. Here we focused on communities of 20 competing species with $\mu_i^0 = 0.5 h^{-1}$, $D = 0.2 h^{-1}$, $\gamma_{ij} = 0.9$. $\mu_i$ values were randomized 5000 times following uniform distributions. Each time the mean of the uniform distribution equaled 0.5, while the distribution width was randomized between 0 and 0.5.

## Quantification of population diversity by Shannon index

Shannon index, a common metric used in ecology to measure diversity, is defined as follows:

$$H = \exp\left[-\sum\left(\frac{s_i}{s_T}\log\frac{s_i}{s_T}\right)\right] \tag{9}$$

where $s_i$ is the abundance of the $i$-th species and $s_T$ is the total abundance: $s_T = \sum_{i=1}^{m} s_i$.

## Reporting summary

Further information on research design is available in the Nature Portfolio Reporting Summary linked to this article.

## Data availability

The simulation data generated in this study have been deposited in the Github repository[72]. Source data are provided with this paper.

## Code availability

All the codes associated with the numerical simulations and analysis of this paper are available at the Github repository[72].

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

## Acknowledgements
We thank Prof. Po-Yi Ho and Prof. Xiao Yi for their comments and suggestions on an earlier draft of this paper. This work is supported by the startup fund awarded to Teng Wang by Shenzhen Institute of Synthetic Biology.

## Author contributions
S.Z. conceived the research, performed the modeling, conducted the analysis, and wrote the paper. J.H. assisted with modeling and manuscript editing. T.W. conceived the research, performed the modeling, conducted the analysis, and wrote the paper.

## Competing interests
The authors declare no competing interests.
