## [Peer Review File · Nature Communications]

Horizontal gene transfer is predicted to overcome the diversity limit of competing microbial speciesREVIEWER COMMENTS

Reviewer #1 (Remarks to the Author):

Summary

In this paper, the authors analyze a modified Lotka-Volterra model incorporating horizontal gene-transfer between species (or strains). They show that horizontal gene transfer can in principal act as a mechanism for the maintenance of diversity by reducing fitness differences between species leading to neutral coexistence. Importantly whereas standard neutral coexistence is unlikely to persist under environmental perturbations, the dynamic nature of horizontal gene transfer can leads to a re-convergence in fitness and and neutral coexistence even in changing environments (“dynamic neutrality”).

This is a thought-provoking and excellently written paper that tackles a long-standing question in microbial population biology and community ecology. The proposed mechanism by which HGT maintains diversity is intuitive, well explained and backed up by rigorous analytical derivations and numerical simulations. The work presented is technically sound and I see no major issues with how the results have been generated. The authors should be commended for the excellent figures and the concise explanation of the core concepts in the main text. I found the analysis presented in Figure 4 to be especially elegant and insightful.

Despite these strengths I have some substantive comments that would need to be addressed if the authors want to justify the main conclusion of the paper, which is that HGT will act as mechanism for maintenance of coexistence in microbial communities. Firstly there are a couple of big assumptions that the authors have not fully acknowledged and which I suspect will limit the extent to which the proposed mechanism plays a significant role in most microbial systems. Secondly, the author have not established that empirically observed levels of HGT are sufficient for the proposed effect to have a more than negligible impact especially when compared to other known mechanisms for coexistence in homogeneous environments. Neither of these issues should preclude eventual publication of the paper which is undoubtedly important for the field but it may require the authors to

do additional work and provide more nuance in the discussion.

Major Comments

1. A major assumption of this paper is that two species will become more similar in fitness when they share the same mobile genetic element. However, the same mobile genetic element can have very different fitness effects in different genetic background due to epistasis (I.e Kirit, Lagator and Bollback 2020). The authors should address how HGT will affect coexistence when mobilizable genes have different effects across genetic background? My intuition is that magnitude epistasis will not qualitatively change the results whereas sign epistasis will.
2. A second major assumption in this paper is that sharing genetic elements leads to more similar growth rates but does not increase the amount of inter-specific competition which would decrease coexistence feasibility (lines 80-82). However, different mobile genetic elements affect different phenotypes, some of which will have larger effect on growth rate parameters and some which will have larger effect on competition parameters. For example the sharing of metabolic genes can lead to overlapping resource usage which would decrease niche partitioning, thus increasing the amount of inter-specific competition.(I.e Caro-Quintero and Konstantinidis 2015, Bonham, Wolfe and Dutton 2017) . The authors could address this through additional simulations that explicitly explore the impact of HGT on coexistence when shared genes increase the strength of competition. I suspect that addressing these nuances may require the authors to tone down the discussion.
3. In the model the level of HGT is set by the transfer rate η . Simulations are performed for values of η evenly spread out along a linear scale between 0 and 1. The authors have not explicitly listed the units for every parameter in the model so its hard to get a sense for how the values of η simulated correspond to empirically observed levels of HGT. My sense is that empirically estimated rates of gene transfer (such as those measured in Kosterlitz et al 2022) are several orders of magnitude lower than those simulated in the model, which would potentially make the described effects on diversity negligible when compared to other mechanisms for coexistence in homogeneous environments such as growth trade-offs or cross-feeding (i.e Manhart and Shakhnovich 2018 , Goldford et al 2018). Under what conditions would the authors expect HGT to play an over-sized role when compared to

these other mechanisms?

4. The discussion needs to better contextualize the results within the broader literature. Specifically, there are a couple of recent papers that have touched on the role of HGT in microbial community coexistence and stability (i.e. Coyte et al 2022 and Fan et al 2018.) . My impression is that the models used in those two papers are more system specific compare to the model the authors have explored, but the discussion would be strengthened by explaining how the mechanisms maintaining diversity compare.

Referenced Papers

Acar Kirit H, Lagator M, Bollback JP. Experimental determination of evolutionary barriers to horizontal gene transfer. *BMC Microbiol.* 2020 Oct 28;20(1):326. doi: 10.1186/s12866-020-01983-5. PMID: 33115402; PMCID: PMC7592521.

Caro-Quintero, A., Konstantinidis, K. Inter-phylum HGT has shaped the metabolism of many mesophilic and anaerobic bacteria. *ISME J* 9, 958–967 (2015).
<https://doi.org/10.1038/ismej.2014.193>

Kevin S Bonham, Benjamin E Wolfe, Rachel J Dutton (2017) .Extensive horizontal gene transfer in cheese-associated bacteria *eLife* 6:e22144.

Kosterlitz, O., A. Muñiz Tirado, C. Wate, C. Elg, I. Bozic, E. M. Top, B. Kerr (2022) Estimating the transfer rates of bacterial plasmids with an adapted Luria–Delbrück fluctuation analysis. *PLoS Biology*

Goldford J, Lu N, Bajic D, Estrela S, Sanchez-Gorostiaga A, Tikhonov M, Segre D, Mehta P, Sanchez A (2018) . Emergent simplicity in microbial community assembly. *Science*

Manhart, M., Shakhnovich, E.I. Growth tradeoffs produce complex microbial communities on a single limiting resource. *Nat Commun* 9, 3214 (2018).

Katharine Z. Coyte ,Cagla Stevenson,Christopher G. Knight,Ellie Harrison,James P. J. Hall,Michael A. Brockhurst Horizontal gene transfer and ecological interactions jointly control microbiome stability, *Plos Biology* (2022).

Reviewer #2 (Remarks to the Author):

Reviewer Summary for the Manuscript titled "Horizontal Gene Transfer Overcomes the Diversity Limit of Competing Microbial Species"

Zhu, Hong, and Wang (2023) address crucial research questions regarding the often-overlooked role of Mobile Genetic Elements (MGEs) in species competition and coexistence. The authors rightly emphasize the dynamic influence of Horizontal Gene Transfer (HGT) on species growth rates through MGE fitness effects, prompting an exploration into the extent to which this aspect is overlooked in studies of species coexistence. They present an innovative modeling framework that incorporates gene flow among competing microbes, employing both theoretical derivations and numerical simulations, which is commendable.

The significance of this study lies in its potential to reshape the field of microbial ecology by shedding light on a previously underestimated mechanism and providing valuable insights for the design and engineering of complex microbial consortia. However, it would be beneficial for the authors to provide additional context within the broader field of ecology and species coexistence research. Below are our major comments.

1. The assertion that contemporary models predominantly consider fitness as a constant concept could be rephrased, as the ecological literature has actively explored dynamic fitness concepts. The authors may want to acknowledge the existing body of work in animals, plants, and microbial interactions where dynamic fitness is explicitly considered. Examples from eco-evolutionary feedback studies (e.g., sticklebacks, guppies, cichlids) and niche construction in microbial interactions (e.g., nutrient uptake and metabolism) could be highlighted to support this point. Additionally, it's crucial to explore whether the dynamic nature of fitness arises from changes in population structure or shifts in ecological environments.

2. Clarification is needed regarding the biological significance of the perturbations introduced in the model, especially in comparison to the perturbations in relative abundance commonly used in stability analysis. This may help determine if the system exhibits alternative stable states.

3. Figure 4 shows the persistence of five species in the absence of MGEs after 600 time steps. It would be valuable to assess if these coexisting species rely on neutrality and to investigate the similarity in their fitness values. Exploring the role of initial variance in fitness within the population could help explain how quickly the system reaches a stable state.

4. We are curious about the model generality. It would be informative to assess whether the results still hold under various scenarios, such as dynamic interspecific coefficients (as opposed to static ones), MGE effects dependent on genetic backgrounds (i.e., epistasis), discrete vs. continuous fitness effects of MGEs (i.e., gain or loss of antibiotic resistance plasmids results in survival or death), and scenarios with a higher diversity of MGEs than bacterial chromosomes (ie, the MGEs insert a high variability than chromosomes do). Addressing how the diversity of MGEs, whether through immigration or de novo mutations, impacts the results could provide valuable insights.

5. The authors could consider referencing previous work showing how MGEs prevent selective sweeps caused by mutations in the presence of immigration (Niehus et al., 2015, Nat Comm) and show analysis or discuss how MGE diversity through gene flow with external populations might affect their results.

Point-by-point responses (in black) to reviewers' comments (in blue).

Reviewer #1 (Remarks to the Author):

Summary:

In this paper, the authors analyze a modified Lotka-Volterra model incorporating horizontal gene-transfer between species (or strains). They show that horizontal gene transfer can in principal act as a mechanism for the maintenance of diversity by reducing fitness differences between species leading to neutral coexistence. Importantly whereas standard neutral coexistence is unlikely to persist under environmental perturbations, the dynamic nature of horizontal gene transfer can lead to a re-convergence in fitness and neutral coexistence even in changing environments (“dynamic neutrality”).

This is a thought-provoking and excellently written paper that tackles a long-standing question in microbial population biology and community ecology. The proposed mechanism by which HGT maintains diversity is intuitive, well explained and backed up by rigorous analytical derivations and numerical simulations. The work presented is technically sound and I see no major issues with how the results have been generated. The authors should be commended for the excellent figures and the concise explanation of the core concepts in the main text. I found the analysis presented in Figure 4 to be especially elegant and insightful. Despite these strengths I have some substantive comments that would need to be addressed if the authors want to justify the main conclusion of the paper, which is that HGT will act as mechanism for maintenance of coexistence in microbial communities. Firstly there are a couple of big assumptions that the authors have not fully acknowledged and which I suspect will limit the extent to which the proposed mechanism plays a significant role in most microbial systems. Secondly, the author have not established that empirically observed levels of HGT are sufficient for the proposed effect to have a more than negligible impact especially when compared to other known mechanisms for coexistence in homogeneous environments. Neither of these issues should preclude eventual publication of the paper which is undoubtedly important for the field but it may require the authors to do additional work and provide more nuance in the discussion.

We appreciate the reviewer for finding this paper interesting, important and well-written. We are also extremely grateful for the reviewer's in-depth comments and suggestions, which are very helpful for us to further justify our main conclusion. We have fully addressed all the issues in the updated manuscript and below.

Major Comments

1. A major assumption of this paper is that two species will become more similar in fitness when they share the same mobile genetic element. However, the same mobile genetic element can have very different fitness effects in different genetic background due to epistasis (I.e Kirit, Lagator and Bollback 2020). The authors should address how HGT will affects coexistence when mobilizable genes have different effects across genetic background? My intuition is that magnitude epistasis will not qualitatively change the results whereas sign epistasis will.

We are grateful for the reviewer's suggestion, which is quite interesting and relevant. Indeed, epistasis of mobile genetic elements is an important aspect that should be considered in our analysis. To explore how HGT will affect coexistence when mobile genes have different effects across genetic backgrounds, we built a two-species competition model that accounts for the two types of epistasis and carried out additional numerical simulations. Our numerical results agree with the reviewer's prediction: magnitude epistasis does not qualitatively change the results whereas sign epistasis does.

The detailed description of the epistasis model was provided in Section 2 of Supplementary Information. The simulation results were shown in Supplementary Fig. 5. We also added a new paragraph in the main text to discuss the influence of epistasis on the interplay between HGT and species coexistence (lines #208-220):

- ‘In the two-species model, we assumed that the metabolic burden or benefit of an MGE was independent of the host species or strains. However, in nature the same MGE can have different fitness effects in different genetic backgrounds due to epistasis¹⁻⁴. To evaluate the influence of this assumption on the conclusion, we built a model that accounted for two types of epistasis: magnitude epistasis, where the host genomic background only influences the magnitude but not the sign of the fitness effect, and sign epistasis, where the same MGE causes growth burden in one species or strain while brings fitness benefit in the other (see Supplementary Information for more details). Our numerical simulations with randomized parameters suggest that how HGT affects coexistence is dependent on the epistasis type. Magnitude epistasis doesn’t qualitatively change the conclusion but sign epistasis does (Supplementary Fig. 5). When a mobile gene causes opposite fitness effects in two different genetic backgrounds, the transfer of this gene will reduce the coexistence feasibility. These results suggest that MGE epistasis might add another layer of complexities into the interplay between HGT and species coexistence.’

We also thank the reviewer for bringing a relevant paper into our attention. We have cited this paper in the main text.

2. A second major assumption in this paper is that sharing genetic elements leads to more similar growth rates but does not increase the amount of inter-specific competition which would decrease coexistence feasibility (lines 80-82). However, different mobile genetic elements affect different phenotypes, some of which will have larger effect on growth rate parameters and some which will have larger effect on competition parameters. For example the sharing of metabolic genes can lead to overlapping resource usage which would decrease niche partitioning, thus increasing the amount of inter-specific competition. (I.e Caro-Quintero and Konstantinidis 2015, Bonham, Wolfe and Dutton 2017) . The authors could address this through additional simulations that explicitly explore the impact of HGT on coexistence when shared genes increase the strength of competition. I suspect that addressing these nuances may require the authors to tone down the discussion.

We appreciate the reviewer for the insightful comments. The reviewer raised a very important point that the transfer of some metabolic genes might increase the strength of inter-species competition. In light of the reviewer’s suggestion, we have carried out additional simulations to analyze the effects of HGT on coexistence when gene transfer promotes competition. Our results suggested that HGT can reduce the coexistence feasibility when mobile genes promote competition.

In the updated Supplementary Information, we added a new section entitled ‘The effect of HGT on species coexistence when mobile genes promote inter-species competition’ to provide the details of the model and simulations. The numerical results were provided in Supplementary Fig. 6. In the updated main text, we added a new paragraph in Discussion to address this issue as a potential caveat of the main conclusion (lines #221-231):

- ‘Our work predicts that HGT promotes species diversity when MGEs only affect species growth rates and have no influences on inter-species competition. Certain caveats need to be considered

when applying this prediction. For instance, the sharing of many mobile genes can also promote niche overlapping, leading to the increase of competition strength^{5,6}. To understand how the transfer of these genes will influence species coexistence, we adapted the main model, by considering the dynamic change of competition strength during gene transfer (see Supplementary Information for more details). The numerical simulations predict that when mobile genes promote inter-species competition, HGT can reduce the coexistence feasibility of competing species (Supplementary Fig. 6). These results suggest that how HGT affects species coexistence in a specific microbiota might be context-dependent. Gene transfer might promote or suppress population diversity, depending on the biological traits encoded by the mobile genes.'

3. In the model the level of HGT is set by the transfer rate η . Simulations are performed for values of η evenly spread out along a linear scale between 0 and 1. The authors have not explicitly listed the units for every parameter in the model so its hard to get a sense for how the values of η simulated correspond to empirically observed levels of HGT. My sense is that empirically estimated rates of gene transfer (such as those measured in Kosterlitz et al 2022) are several orders of magnitude lower than those simulated in the model, which would potentially make the described effects on diversity negligible when compared to other mechanisms for coexistence in homogeneous environments such as growth trade-offs or cross-feeding (i.e. Manhart and Shakhnovich 2018 , Goldford et al 2018). Under what conditions would the authors expect HGT to play an over-sized role when compared to these other mechanisms?

We thank the reviewer for the important comments, which are very helpful for us to specify the context where our predictions are most applicable. We apologize for the lack of clarification on the parameter units. In section 1 of the updated Supplementary Information, we have listed the unit of every parameter and variable in the model. We also clearly labeled the parameter units in every figure and its legend.

The unit of η in our model is $hour^{-1}$, which is different from the unit of empirically measured transfer rates (denoted as η^c , with the unit of $cells^{-1} \cdot mL \cdot hour^{-1}$). The difference comes from the process of non-dimensionalizing population abundances (see section 1 of Supplementary Information). Our η is related with the empirical η^c by $\eta = \eta^c \cdot N_m$, where N_m is the maximum carrying capacity of the population. N_m , with the unit of $cells \cdot mL^{-1}$, is typically large. For instance, in LB culture⁷, N_m of an *E. coli* population can reach $10^9 cells \cdot mL^{-1}$. In human colon⁸, N_m can reach about $10^{12} cells \cdot mL^{-1}$. This explains why the empirically estimated rates of gene transfer were several orders of magnitude lower than those simulated in our model. When N_m is sufficiently large, even slight gene transfer rate η^c can have significant effects on species coexistence feasibility. This intuition was confirmed by our additional numerical simulations as shown in Supplementary Fig. 4a.

We also thank the reviewer for recommending a highly relevant paper (Kosterlitz et al 2022). In this paper, the authors estimated the transfer rates of a conjugative plasmid cross species (from *Klebsiella pneumoniae* to *Escherichia coli*) and within species (between *E. coli* strains). Using the LDM method that they established, the cross-species and within-species conjugation rates were estimated to be around 10^{-13} and $10^{-7} cells^{-1} \cdot mL \cdot hour^{-1}$, respectively. In our Supplementary Information, we explicitly examined whether these two empirical estimates are sufficient for the proposed effects. As shown in Supplementary Fig. 4b and c, our results suggested that with $\eta^c = 10^{-7} cells^{-1} \cdot mL \cdot hour^{-1}$, HGT is able to significantly promote coexistence in many environments such as ocean ($10^6 \sim 10^8$ cells per mL)⁹ or river waters (some can reach $\sim 8 \times 10^6$ cells per mL)¹⁰. Even with $\eta^c = 10^{-13} cells^{-1} \cdot mL \cdot hour^{-1}$, HGT is able to significantly promote species coexistence in some natural environments like human colon.

We also tested the influence of MGE fitness effects. Our numerical simulations suggest that when MGEs are beneficial, extremely small HGT rates can significantly promote species coexistence (Supplementary Fig. 4d).

The details of the parameter units, model establishments and supplemented simulations were provided in section 1 of the Supplementary Information. We summarized the simulation results in Supplementary Fig. 4. We also added a new paragraph in main text to compare our η with the empirical measurements and discuss the conditions where HGT plays non-negligible role (lines #180-195):

- ' The empirically estimated gene transfer rates (denoted as η^c) need to be multiplied by N_m before being plugged into our model (see Supplementary Information for more details). Here N_m is the maximum carrying capacity of the population and has the unit of cells per mL. Therefore, the transfer rates η in our model are several orders of magnitude higher than those measured in previous studies^{7,11}. When N_m is large, even slight η^c can significantly change the coexistence feasibility. In contrast, with small N_m , the effects of η^c can become negligible (Supplementary Fig. 4a). Using two empirical estimates of plasmid conjugation rates from a previous study¹¹, our numerical simulations suggest that the empirical HGT rates are sufficient to promote coexistence in a wide range of natural conditions^{8,11} (Supplementary Fig. 4b and c). We also explored the influence of MGE fitness effects on the effective range of transfer rates. Our simulations show that when MGEs are beneficial, extremely small HGT rates can be effective to promote diversity (Supplementary Fig. 4d, see Supplementary Information for more details). These results suggest that the role of HGT may become prominent in many environments especially those with high cell density and beneficial MGEs. However, for burdensome MGEs in environments with very low cell densities, the contribution of HGT can be less important than other mechanisms like growth tradeoff or cross-feeding^{12,13}.

4. The discussion needs to better contextualize the results within the broader literature. Specifically, there are a couple of recent papers that have touched on the role of HGT in microbial community coexistence and stability (i.e. Coyte et al 2022 and Fan et al 2018.) . My impression is that the models used in those two papers are more system specific compare to the model the authors have explored, but the discussion would be strengthened by explaining how the mechanisms maintaining diversity compare.

We sincerely thank the reviewer for the suggestion. We are also grateful for the two reference papers, which are very helpful for us to relate this work with the broad literature. In light of the reviewer's suggestion, we have added a new paragraph in Discussion to compare our work with the previous studies (lines #232-238):

- 'Our results are in line with the previous studies on the relationship between HGT and microbiota stability^{14,15}. For instance, one study focused on the interaction between microbial cooperators and cheaters, and showed that HGT could promote the coexistence of these two genotypes¹⁴. Another study specifically showed that horizontal transfer of resistance genes could promote the microbiome stability in response to environmental stressors¹⁵. While these studies only considered specific systems, our work resonates with their conclusions and generalizes the role of HGT into broader context of microbial communities.'

Referenced Papers

Acar Kirit H, Lagator M, Bollback JP. Experimental determination of evolutionary barriers to horizontal gene transfer. *BMC Microbiol.* 2020 Oct 28;20(1):326. doi: 10.1186/s12866-020-01983-5. PMID: 33115402; PMCID: PMC7592521.

Caro-Quintero, A., Konstantinidis, K. Inter-phylum HGT has shaped the metabolism of many mesophilic and anaerobic bacteria. *ISME J* 9, 958–967 (2015). <https://doi.org/10.1038/ismej.2014.193>

Kevin S Bonham, Benjamin E Wolfe, Rachel J Dutton (2017) .Extensive horizontal gene transfer in cheese-associated bacteria *eLife* 6:e22144.

Kosterlitz, O., A. Muñoz Tirado, C. Wate, C. Elg, I. Bozic, E. M. Top, B. Kerr (2022) Estimating the transfer rates of bacterial plasmids with an adapted Luria–Delbrück fluctuation analysis. *PLoS Biology*

Goldford J, Lu N, Bajic D, Estrela S, Sanchez-Gorostiaga A, Tikhonov M, Segre D, Mehta P, Sanchez A (2018) . Emergent simplicity in microbial community assembly. *Science*

Manhart, M., Shakhnovich, E.I. Growth tradeoffs produce complex microbial communities on a single limiting resource. *Nat Commun* 9, 3214 (2018).

Katharine Z. Coyte ,Cagla Stevenson,Christopher G. Knight,Ellie Harrison,James P. J. Hall,Michael A. Brockhurst. Horizontal gene transfer and ecological interactions jointly control microbiome stability, *Plos Biology* (2022).

We appreciate the reviewer for generously recommending many important papers. We have cited and discussed these papers in the updated manuscript.

Reviewer #2 (Remarks to the Author):

Reviewer Summary for the Manuscript titled "Horizontal Gene Transfer Overcomes the Diversity Limit of Competing Microbial Species"

Zhu, Hong, and Wang (2023) address crucial research questions regarding the often-overlooked role of Mobile Genetic Elements (MGEs) in species competition and coexistence. The authors rightly emphasize the dynamic influence of Horizontal Gene Transfer (HGT) on species growth rates through MGE fitness effects, prompting an exploration into the extent to which this aspect is overlooked in studies of species coexistence. They present an innovative modeling framework that incorporates gene flow among competing microbes, employing both theoretical derivations and numerical simulations, which is commendable.

The significance of this study lies in its potential to reshape the field of microbial ecology by shedding light on a previously underestimated mechanism and providing valuable insights for the design and engineering of complex microbial consortia. However, it would be beneficial for the authors to provide additional context within the broader field of ecology and species coexistence research. Below are our major comments.

We appreciate the reviewer's positive comments on the value and novelty of this work. We are also extremely grateful for the reviewer's suggestions on relating our work with broader literature. These suggestions are very helpful to further improve the quality and rigor of this manuscript. All the comments have been addressed in the updated manuscript and below.

1. The assertion that contemporary models predominantly consider fitness as a constant concept could be rephrased, as the ecological literature has actively explored dynamic fitness concepts. The authors may want to acknowledge the existing body of work in animals, plants, and microbial interactions where dynamic fitness is explicitly considered. Examples from eco-evolutionary feedback studies (e.g., sticklebacks, guppies, cichlids) and niche construction in microbial interactions (e.g., nutrient uptake and metabolism) could be highlighted to support this point. Additionally, it's crucial to explore whether the dynamic nature of fitness arises from changes in population structure or shifts in ecological environments.

We are grateful for the reviewer's insightful comments. In the updated manuscript, we have avoided the assertion 'most models assumed that the species fitness was temporally constant' to more accurately reflect the literature. Throughout the main text and Supplementary Information, we have rephrased or deleted any similar statements whenever applicable. To acknowledge the existing studies on dynamic fitness, we have added a paragraph in Introduction as follows (lines #45-52):

- 'Many studies have documented the rapid evolutionary change of species fitness on ecological timescale, which leads to the emergence of eco-evolutionary dynamics¹⁶⁻²⁰. For instance, in sticklebacks^{16,17}, guppies¹⁸ and cichlids^{19,20}, the eco-evolutionary feedbacks have been widely observed. A microbial species can also actively change its niche by modifying its nutrient uptake and metabolism^{21,22}. The dynamic change of species fitness can have significant influence on the ecological outcomes, including species coexistence. HGT is a key mechanism that mediates the eco-evolutionary interplay in microbes²³. However, a theoretical framework allowing the analysis of HGT's effects on microbial diversity remains lacking.'

The reviewer also raised a critical point that the dynamic fitness can emerge from changes in population structure or shifts in ecological environments. We agree with the reviewer that it is important to distinguish

these two mechanisms. In light of the reviewer's comments, in the Discussion of the updated manuscript we have added (lines #239-245):

- 'In our extended LV model, the dynamics of species fitness during HGT arises from the changes in population structure: in each species, HGT generates subpopulations whose growth rates differ from the others due to the metabolic burden or benefits of the mobile genes. However, the dynamic fitness can also emerge from the shifts of environmental factors²⁴⁻²⁶. The influence of environmental changes on species coexistence has been extensively studied²⁷⁻²⁹. Together with these previous studies, our work highlights the importance of considering the dynamic nature of species fitness in the field of ecology.'

2. Clarification is needed regarding the biological significance of the perturbations introduced in the model, especially in comparison to the perturbations in relative abundance commonly used in stability analysis. This may help determine if the system exhibits alternative stable states.

We thank the reviewer for the comments. We have added a paragraph in Discussion of the updated main text to clarify the biological significance of the perturbations introduced in the model and compare the two types of perturbations as follows (lines #196-207):

- 'Natural microbial communities are often faced with constantly varying environmental conditions which affect the ecological parameters such as species growth rates. In order for the diversity to be stably maintained, the population needs to be insensitive to the perturbations of the parameters³⁰⁻³². Mathematically such robustness translates into coexistence feasibility or structural stability, which measures the volume of the parameter space that allows the positive abundances of all species³⁰⁻³³. We noted that the structural stability, by definition, is different from the dynamical stability or local asymptotic stability, which refers to the ability of a system to recover after perturbation in species relative abundances^{30,31}. While the local asymptotic stability has been extensively studied in many systems³⁴⁻³⁶, the determinants of structural stability have been less understood. In this work, we explicitly showed that HGT can promote the structural stability of microbial communities. How HGT influences the local asymptotic stability remains an open question for future studies.'

We also examined the number of alternative stable states by performing additional numerical simulations. Our results suggested that the number of alternative stable states increases with HGT rate, indicating the role of gene transfer on the global stability landscape of microbial communities. We provided the details of the numerical simulations in section 4 of the Supplementary Information. The simulation results were presented in Supplementary Fig. 7.

3. Figure 4 shows the persistence of five species in the absence of MGEs after 600 time steps. It would be valuable to assess if these coexisting species rely on neutrality and to investigate the similarity in their fitness values. Exploring the role of initial variance in fitness within the population could help explain how quickly the system reaches a stable state.

We are extremely grateful for the reviewer's suggestions. To analyze whether the species coexistence after random perturbations relies on neutrality, we repeated the simulation for 2000 times. In each simulation, we calculated the similarity in species fitness values and population diversity. Our results suggested that the diversity is positively associated with neutrality. In other words, these coexisting species rely on neutrality. There will be more species coexisting when their growth rates are similar after perturbations. The results were provided in Supplementary Fig. 8a.

To analyze the role of initial variance in fitness, we performed additional simulations by randomizing initial neutrality. Specially, at time zero, we randomized the species growth rates around 0.5 following uniform distributions. The initial neutrality is reflected by the distribution width: smaller width stands for greater initial neutrality. After time zero, perturbations were introduced into species growth rates at random intervals. We then calculated the time point τ when the community diversity reduced by 50%. τ reflects how quickly the systems will reach a stable state. Our results suggested that τ decreases with initial variance in species fitness. The diversity declines more quickly when the initial fitness variance is high. These results were presented in Supplementary Fig. 8b and c. The details of the numerical simulations were provided in the figure legend.

4. We are curious about the model generality. It would be informative to assess whether the results still hold under various scenarios, such as dynamic interspecific coefficients (as opposed to static ones), MGE effects dependent on genetic backgrounds (i.e., epistasis), discrete vs. continuous fitness effects of MGEs (i.e., gain or loss of antibiotic resistance plasmids results in survival or death), and scenarios with a higher diversity of MGEs than bacterial chromosomes (ie, the MGEs insert a high variability than chromosomes do). Addressing how the diversity of MGEs, whether through immigration or de novo mutations, impacts the results could provide valuable insights.

We thank the reviewer for the suggestions. For each scenario that the reviewer suggested, we have performed additional simulations and analysis to examine the generality of our model.

(1) For dynamic interspecies competition, indeed the sharing of many mobile genes can promote niche overlapping, leading to the increase of competition strength. We analyzed the effects of HGT on coexistence when gene transfer promotes competition. Our results suggest that HGT can reduce the coexistence feasibility when mobile genes increase the competition strength. In the updated Supplementary Information, we added a new section entitled ‘The effect of HGT on species coexistence when mobile genes promote inter-species competition’ to provide the details of the modeling framework and simulation pipelines. The simulation results are summarized in Supplementary Figure 6. In the updated main text, we added a new paragraph in Discussion to address the potential caveats of the main conclusion (lines #221-231):

- ‘Our work predicts that HGT promotes species diversity when MGEs only affect species growth rates and have no influences on inter-species competition. Certain caveats need to be considered when applying this prediction. For instance, the sharing of many mobile genes can also promote niche overlapping, leading to the increase of competition strength^{5,6}. To understand how the transfer of these genes will influence species coexistence, we adapted the main model, by considering the dynamic change of competition strength during gene transfer (see Supplementary Information for more details). The numerical simulations predict that when mobile genes promote inter-species competition, HGT can reduce the coexistence feasibility of competing species (Supplementary Fig. 6). These results suggest that how HGT affects species coexistence in a specific microbiota might be context-dependent. Gene transfer might promote or suppress population diversity, depending on the biological traits encoded by the mobile genes.’

(2) In the scenario where MGE’s effects depend on genetic backgrounds, we built a two-species competition model that took epistasis into account. We tested two types of epistasis: magnitude epistasis, where the host genomic background only influences the magnitude but not the sign of the fitness effect, and sign epistasis, where the same mobile genetic element causes growth burden in one species while brings

fitness benefit in the other. Our numerical results suggest that magnitude epistasis does not qualitatively change the result whereas sign epistasis does. We have added a new section entitled ‘Epistasis of mobile genetic elements’ in Supplementary Information to provide the details of the epistasis model. The numerical results were presented in Supplementary Fig. 5. In the updated main text, we added a paragraph in Discussion to discuss the influence of epistasis on the interplay between HGT and species coexistence (lines #208-220):

- ‘In the two-species model, we assumed that the metabolic burden or benefit of an MGE was independent of the host species or strains. However, in nature the same MGE can have different fitness effects in different genetic backgrounds due to epistasis¹⁻⁴. To evaluate the influence of this assumption on the conclusion, we built a model that accounted for two types of epistasis: magnitude epistasis, where the host genomic background only influences the magnitude but not the sign of the fitness effect, and sign epistasis, where the same MGE causes growth burden in one species while brings fitness benefit in the other (see Supplementary Information for more details). Our numerical simulations with randomized parameters suggest that how HGT affects coexistence is dependent on the epistasis type. Magnitude epistasis doesn’t qualitatively change the conclusion but sign epistasis does (Supplementary Fig. 5). When a mobile gene causes opposite fitness effects on two different species, the transfer of this gene will reduce their coexistence feasibility. These results suggested that MGE epistasis might add another layer of complexities into the interplay between HGT and species coexistence.’

(3) For discrete fitness effects of MGEs, we explored the species diversity with or without HGT under strong antibiotic selection. Specifically, we considered the transfer of an antibiotic resistant MGE in a population. Only cells carrying the MGE can survive. Our numerical simulations suggest that increasing HGT rate promotes the species coexistence and diversity. We provided the details of the model and simulations in a new section entitled ‘Discrete fitness effects of MGEs’ in Supplementary Information. The simulation results were presented in Supplementary Fig. 9. We further discussed the discrete fitness effects of MGEs in Discussion of the updated main text as follows (lines #246-253):

- ‘The fitness effect of an MGE can be discrete. For instance, under strong antibiotic selections, only cells carrying the antibiotic resistant MGEs can survive. To examine whether our conclusion is still applicable in this scenario, we generalized our model by considering the transfer of an antibiotic resistant MGE in a population of m species (see Supplementary Information for more details). Our numerical result suggests that without HGT, only the donor species carrying the MGE can survive due to the antibiotic selection. Increasing HGT rate promotes the species coexistence and diversity by allowing more species to be resistant to antibiotic killing (Supplementary Fig. 9). These results suggest the applicability of our conclusion to the scenario of discrete fitness effects.’

(4) The reviewer also brings up the diversity of MGEs. Indeed, in our model, we assumed the number of the MGEs equaled the species number. To understand whether our conclusion is still applicable when the diversity of MGEs is higher than bacterial chromosomes, we established an extended model which considered the flow of an arbitrary number (denoted as n) of MGEs in a community of m species. By numerical simulations with randomized parameters, our results suggest that increasing MGE diversity promotes the coexistence feasibility. In addition, regardless of the MGE diversities, coexistence feasibilities can always be promoted by increasing HGT rate. The details of the model and simulations were provided

in a new section entitled ‘The diversity of MGEs’ of Supplementary Information. The simulation results were provided in Supplementary Fig. 10. In the updated manuscript, we added one paragraph to discuss the influence of MGE diversity as follows (lines #254-263):

- ‘In our model, we assumed the number of the MGEs equaled the species number. However, in natural systems, the MGE diversity might be higher than chromosomes due to immigration or de novo mutations³⁷. To understand whether our conclusion is still applicable when the diversity of MGEs changes, we established an extended model which accounts for the flow of an arbitrary number of MGEs in a community of multiple species. By numerical simulations with randomized parameters, our results show that the coexistence feasibility increases with MGE diversity (Supplementary Fig. 10). In addition, regardless of the MGE diversity, faster HGT rate consistently leads to greater possibility of coexistence. These results suggest that enhancing genetic exchange among microbes, through either increasing MGE diversity or increasing HGT rate, can promote the microbial diversity.’

5. The authors could consider referencing previous work showing how MGEs prevent selective sweeps caused by mutations in the presence of immigration (Niehus *et al.*, 2015, *Nat Comm*) and show analysis or discuss how MGE diversity through gene flow with external populations might affect their results.

We thank for the reviewer for the helpful suggestion. Indeed, the work by Niehus *et al.* is very interesting and relevant, showing how horizontal gene transfer prevents vertical selective sweeps when migration is present. Our work shows that HGT can promote the coexistence of strong and weak competitors, reducing the likelihood of winner-taking-all scenarios while allowing the mobile genes to spread across species, which is in line with their results. Our results also suggest the role of HGT can be amplified by greater MGE diversity. Therefore, MGE diversity through gene flow with external populations can further promotes the frequency of horizontal sweeps relative to vertical sweeps. Such MGE diversity might allow the horizontal sweeps of alleles under positive selections even in the absence of immigrations. We have added a paragraph in Discussion of the updated main text to thoroughly discuss how our conclusions might affect their results (lines #270-277).

Reference

- 1 Acar Kirit, H., Lagator, M. & Bollback, J. P. Experimental determination of evolutionary barriers to horizontal gene transfer. *BMC Microbiology* **20**, 1-13 (2020).
- 2 Gama, J. A., Zilhão, R. & Dionisio, F. Plasmid interactions can improve plasmid persistence in bacterial populations. *Frontiers in Microbiology* **11**, 2033 (2020).
- 3 Silva, R. F. *et al.* Pervasive sign epistasis between conjugative plasmids and drug-resistance chromosomal mutations. *PLoS Genetics* **7**, e1002181 (2011).
- 4 San Millan, A., Heilbron, K. & MacLean, R. C. Positive epistasis between co-infecting plasmids promotes plasmid survival in bacterial populations. *The ISME Journal* **8**, 601-612 (2014).
- 5 Bonham, K. S., Wolfe, B. E. & Dutton, R. J. Extensive horizontal gene transfer in cheese-associated bacteria. *Elife* **6**, e22144 (2017).
- 6 Caro-Quintero, A. & Konstantinidis, K. T. Inter-phylum HGT has shaped the metabolism of many mesophilic and anaerobic bacteria. *The ISME Journal* **9**, 958-967 (2015).
- 7 Lopatkin, A. J. *et al.* Persistence and reversal of plasmid-mediated antibiotic resistance. *Nature Communications* **8**, 1689 (2017).

- 8 Dieterich, W., Schink, M. & Zopf, Y. Microbiota in the gastrointestinal tract. *Medical Sciences* **6**, 116 (2018).
- 9 Flemming, H.-C. & Wuertz, S. Bacteria and archaea on Earth and their abundance in biofilms. *Nature Reviews Microbiology* **17**, 247-260 (2019).
- 10 Tiquia, S. Extracellular hydrolytic enzyme activities of the heterotrophic microbial communities of the Rouge River: an approach to evaluate ecosystem response to urbanization. *Microbial Ecology* **62**, 679-689 (2011).
- 11 Kosterlitz, O. *et al.* Estimating the transfer rates of bacterial plasmids with an adapted Luria–Delbrück fluctuation analysis. *PLoS Biology* **20**, e3001732 (2022).
- 12 Manhart, M. & Shakhnovich, E. I. Growth tradeoffs produce complex microbial communities on a single limiting resource. *Nature Communications* **9**, 3214 (2018).
- 13 Goldford, J. E. *et al.* Emergent simplicity in microbial community assembly. *Science* **361**, 469-474 (2018).
- 14 Fan, Y., Xiao, Y., Momeni, B. & Liu, Y.-Y. Horizontal gene transfer can help maintain the equilibrium of microbial communities. *Journal of Theoretical Biology* **454**, 53-59 (2018).
- 15 Coyte, K. Z. *et al.* Horizontal gene transfer and ecological interactions jointly control microbiome stability. *PLoS Biology* **20**, e3001847 (2022).
- 16 Best, R. J. *et al.* Transgenerational selection driven by divergent ecological impacts of hybridizing lineages. *Nature Ecology and Evolution* **1**, 1757-1765 (2017).
- 17 Harmon, L. J. *et al.* Evolutionary diversification in stickleback affects ecosystem functioning. *Nature* **458**, 1167-1170 (2009).
- 18 Reznick, D. N. & Travis, J. Experimental studies of evolution and eco-evo dynamics in guppies (*Poecilia reticulata*). *Annual Review of Ecology, Evolution, and Systematics* **50**, 335-354 (2019).
- 19 Ronco, F. *et al.* Drivers and dynamics of a massive adaptive radiation in cichlid fishes. *Nature* **589**, 76-81 (2021).
- 20 El Taher, A. *et al.* Gene expression dynamics during rapid organismal diversification in African cichlid fishes. *Nature Ecology and Evolution* **5**, 243-250 (2021).
- 21 Laland, K. N., Odling-Smee, F. J. & Feldman, M. W. Evolutionary consequences of niche construction and their implications for ecology. *Proceedings of the National Academy of Sciences* **96**, 10242-10247 (1999).
- 22 Laland, K., Matthews, B. & Feldman, M. W. An introduction to niche construction theory. *Evolutionary Ecology* **30**, 191-202 (2016).
- 23 Arnold, B. J., Huang, I.-T. & Hanage, W. P. Horizontal gene transfer and adaptive evolution in bacteria. *Nature Reviews Microbiology* **20**, 206-218 (2022).
- 24 Sæther, B.-E. & Engen, S. The concept of fitness in fluctuating environments. *Trends in Ecology & Evolution* **30**, 273-281 (2015).
- 25 Nguyen, J., Lara-Gutiérrez, J. & Stocker, R. Environmental fluctuations and their effects on microbial communities, populations and individuals. *FEMS Microbiology Reviews* **45**, fuaa068 (2021).
- 26 Rodríguez - Verdugo, A., Vulin, C. & Ackermann, M. The rate of environmental fluctuations shapes ecological dynamics in a two - species microbial system. *Ecology Letters* **22**, 838-846 (2019).
- 27 Letten, A. D., Dhimi, M. K., Ke, P.-J. & Fukami, T. Species coexistence through simultaneous fluctuation-dependent mechanisms. *Proceedings of the National Academy of Sciences* **115**, 6745-6750 (2018).

- 28 Johnson, E. C. & Hastings, A. Coexistence in spatiotemporally fluctuating environments. *Theoretical Ecology*, 1-34 (2023).
- 29 Liu, M., Rubenstein, D. R., Cheong, S. A. & Shen, S.-F. Antagonistic effects of long-and short-term environmental variation on species coexistence. *Proceedings of the Royal Society B* **288**, 20211491 (2021).
- 30 Grilli, J. *et al.* Feasibility and coexistence of large ecological communities. *Nature communications* **8**, 14389 (2017).
- 31 Saavedra, S., Rohr, R. P., Olesen, J. M. & Bascompte, J. Nested species interactions promote feasibility over stability during the assembly of a pollinator community. *Ecology and Evolution* **6**, 997-1007 (2016).
- 32 Song, C., Rohr, R. P. & Saavedra, S. A guideline to study the feasibility domain of multi-trophic and changing ecological communities. *Journal of Theoretical Biology* **450**, 30-36 (2018).
- 33 Rohr, R. P., Saavedra, S. & Bascompte, J. On the structural stability of mutualistic systems. *Science* **345**, 1253497 (2014).
- 34 May, R. M. Will a large complex system be stable? *Nature* **238**, 413-414 (1972).
- 35 Allesina, S. & Tang, S. Stability criteria for complex ecosystems. *Nature* **483**, 205-208 (2012).
- 36 Allesina, S. *et al.* Predicting the stability of large structured food webs. *Nature Communications* **6**, 7842 (2015).
- 37 Springael, D. & Top, E. M. Horizontal gene transfer and microbial adaptation to xenobiotics: new types of mobile genetic elements and lessons from ecological studies. *Trends in Microbiology* **12**, 53-58 (2004).

REVIEWERS' COMMENTS

Reviewer #1 (Remarks to the Author):

The authors have done an excellent job of addressing my comments and the additional analysis has greatly strengthened the paper. I was especially intrigued by the result that HGT may play a larger role in maintaining coexistence in environments with cell densities on the order of magnitude of the gut microbiome (Figures S4). The new results shown in Figure S5 and S6 are very clear to me and add important nuance. The extent to which HGT will act more like figures S5A-B versus S5C-D/S6 is an important empirical question for the field, and I think the paper does an excellent job establishing the range of possible outcomes. Given the new results the authors have presented, in the final version of the manuscript I would urge the authors to be careful not to over-sell the claim that the model shows that HGT will lead to more coexistence without establishing the conditions under which this result holds (such as that HGT does not increase inter-specific competition). In my view the strength of this manuscript is not that it shows that HGT generically leads to more coexistence but rather than it establishes a novel ecological mechanism for maintaining coexistence via HGT (dynamic neutrality) and maps out the conditions under which this mechanism will play an important role. I am happy to leave any outstanding points to the editor.

Reviewer #2 (Remarks to the Author):

We appreciate the detailed and thorough responses and simulations provided by the authors in addressing the comments raised in our initial review. The authors have done an excellent job addressing our main concerns regarding the model's generality and contextualization. The revisions made to the manuscript are generally satisfactory, and we are pleased to see the improvements that have been implemented.

Regarding some minor comments on the figures:

- The overlapping text on figure objects is challenging to read; for instance, "Diversity limit without HGT" in figure 2b is on top of two lines.
- While we applaud the authors for their figure aesthetics, the color palette is not friendly to color-blinded readers and is not distinguishable in black-and-white. The authors might

consider using another color palette, but we are open to the authors' and editor's decision.

As a disclaimer, we thoroughly checked the manuscript concerning notation and figure references. Although we did not examine the supplementary information in detail, aside from aesthetic decisions, we are pleased to recommend the acceptance of this manuscript to Nature Communications.

Point-by-point responses (in black) to reviewers' comments (in blue).

Reviewer #1 (Remarks to the Author):

The authors have done an excellent job of addressing my comments and the additional analysis has greatly strengthened the paper. I was especially intrigued by the result that HGT may play a larger role in maintaining coexistence in environments with cell densities on the order of magnitude of the gut microbiome (Figures S4). The new results shown in Figure S5 and S6 are very clear to me and add important nuance. The extent to which HGT will act more like figures S5A-B versus S5C-D/S6 is an important empirical question for the field, and I think the paper does an excellent job establishing the range of possible outcomes. Given the new results the authors have presented, in the final version of the manuscript I would urge the authors to be careful not to over-sell the claim that the model shows that HGT will lead to more coexistence without establishing the conditions under which this result holds (such as that HGT does not increase inter-specific competition). In my view the strength of this manuscript is not that it shows that HGT generically leads to more coexistence but rather that it establishes a novel ecological mechanism for maintaining coexistence via HGT (dynamic neutrality) and maps out the conditions under which this mechanism will play an important role. I am happy to leave any outstanding points to the editor.

We appreciate the reviewer for agreeing that our new results have addressed the previous comments and strengthened the paper. We are also grateful for the reviewer's suggestion on toning down the claim that HGT will lead to more coexistence. This suggestion is extremely important for us to more clearly define the novelty of this work. In light of the reviewer's comment, we have thoroughly checked the main text to avoid the overstatement. In particular, we have changed our title to 'Horizontal gene transfer is predicted to overcome the diversity limit of competing microbial species' as the editor suggested. In discussion, we have also clarified: 'Our work proposed an ecological mechanism of maintaining microbial diversity via gene transfer and showed the conditions where this mechanism will potentially be effective.' In addition, we emphasized: 'Gene transfer can promote or suppress microbial coexistence, depending on epistasis and biological trait encoded by the mobile genes.'

Reviewer #2 (Remarks to the Author):

We appreciate the detailed and thorough responses and simulations provided by the authors in addressing the comments raised in our initial review. The authors have done an excellent job addressing our main concerns regarding the model's generality and contextualization. The revisions made to the manuscript are generally satisfactory, and we are pleased to see the improvements that have been implemented.

We thank the reviewer for agreeing that our revisions have addressed the previous comments and improved the quality of this manuscript. We are also grateful for the additional minor comments on the figures. All the issues have been fully addressed in the finalized manuscript and the responses below.

Regarding some minor comments on the figures:

- The overlapping text on figure objects is challenging to read; for instance, "Diversity limit without HGT" in figure 2b is on top of two lines.

We thank the reviewer for pointing it out. We have updated Fig. 2b and Fig. 4b to avoid the overlapping text on figure objects.

- While we applaud the authors for their figure aesthetics, the color palette is not friendly to color-blinded readers and is not distinguishable in black-and-white. The authors might consider using another color palette, but we are open to the authors' and editor's decision.

We appreciated the reviewer for the suggestion. In the updated Fig. 1d, Fig. 2b, Fig. 2c, Fig. 4d, Fig. S4-6, Fig. S9-10, we have used another color palette that is accessible to colorblind people and distinguishable in black-and-white.

As a disclaimer, we thoroughly checked the manuscript concerning notation and figure references. Although we did not examine the supplementary information in detail, aside from aesthetic decisions, we are pleased to recommend the acceptance of this manuscript to Nature Communications.

We thank the reviewer for recommending the acceptance of this manuscript.